# Post-Discharge *Clostridioides difficile* Infection after Arthroplasties in Poland, Infection Prevention and Control as the Key Element of Prevention of *C. difficile* Infections

**DOI:** 10.3390/ijerph19063155

**Published:** 2022-03-08

**Authors:** Estera Jachowicz, Agnieszka Pac, Anna Różańska, Barbara Gryglewska, Jadwiga Wojkowska-Mach

**Affiliations:** 1Department of Microbiology, Faculty of Medicine, Medical College, Jagiellonian University, 31-121 Krakow, Poland; estera.jachowicz@doctoral.uj.edu.pl (E.J.); a.rozanska@uj.edu.pl (A.R.); 2Department of Epidemiology, Medical College, Jagiellonian University, 31-034 Krakow, Poland; agnieszka.pac@uj.edu.pl; 3Department of Internal Medicine and Gerontology, Medical College, Jagiellonian University, 31-501 Krakow, Poland; barbara.gryglewska@uj.edu.pl

**Keywords:** *Clostridioides difficile*, post-discharge surveillance, hip and knee arthroplasties, healthcare-associated CDI

## Abstract

*Clostridioides difficile* is still one of the most common causes of hospital-acquired infectious diarrhea (CDI), and the incidence of CDI is one of the indicators that allows conclusions to be derived on the correctness of antibiotic administration. The objective of this observational study was the analysis of post-discharge CDI incidence in patients undergoing hip or knee arthroplasty, in order to specify optimum conditions for the surgical procedures and outpatient postoperative care. One-year observational study. Public Polish hospitals. Retrospective records for 83,525 surgery patients having undergone hip or knee arthroplasty were extracted from the Polish National Health Fund databases. CDI and/or antibiotic prescriptions in the 30 day post-surgery period were expressed per 1000 surgeries with antibiotic prescription on discharge or in ambulatory care, respectively. The CDI incidence rate was 34.4 per 10,000 patients, and 7.7 cases per 100,000 post-surgery patient-days. Patients who were prescribed at least one antibiotic were diagnosed with CDI more often than patients who had no antibiotic treatment (55.0/1000 patients vs. 1.8/1000 patients). In the multifactorial analysis, the following factors were significant: being at least 65 years of age, trauma as the cause of surgery, length of stay over 7 days, HAIs other than CDI and taking beta-lactams and/or quinolones but not macrolides in the post-discharge period. Postoperative antibiotic prescription in patients undergoing joint replacement surgery is the main risk factor for CDI. These observations indicate the necessity of improvement of infection control programs as the key factor for CDI prevention.

## 1. Introduction

*Clostridioides difficile* is the most frequent cause of nosocomial diarrhea syndrome with possible severe progression. Recurrence of the disease is associated with higher health system costs, as well as exposing patients to additional health risks [1]. CDI (*Clostridioides difficile* infection) is usually associated with healthcare; however, the number of community-acquired infections is on the rise. CDIs are most frequently related to changes in the gut microbiota caused by the administration of antibiotics, and patients who are elderly, immunocompromised or have undergone surgery are at a greater risk of developing CDI and recurrent infection [2].

Healthcare-associated infections, considered as some of the most important public health problems, can be effectively prevented by implementing evidence-based methods of infection prevention and control. Effective implementation also involves monitoring of process and outcomes indicators. The incidence of CDI is one of the indicators that allows conclusions to be derived about the correctness of antibiotic administration, which is also closely related to the prevention of infections. In Poland, there is no network of infection surveillance that would bring together many centers, and research in the area of epidemiology of infections is carried out primarily in single centers. As a result, there are deficiencies in the practice of infection control, e.g., no post-discharge registration, ineffective cooperation between clinicians and microbiological laboratories [3,4] or suboptimal use of antibiotics [5]. According to the European Centre for Disease Prevention and Control (ECDC), the CDI incidence in Poland is almost three times higher than the EU average [6]. In their previous work, the authors have noted the correlations between antibiotic prophylaxis in urology and CDI [5]. The aim of this study is to find out whether similar correlations can be observed in orthopedics.

Data covering large groups of patients include, for example, registers and databases of the public payer for health services in Poland. Analyses of data from the Polish National Health Fund (NHF) databases, taking into account specific groups of patients or areas of public health, may constitute a valuable supplement to the results from single-center studies. The research question of our observational, non-interventional study was: what is the incidence of post-discharge CDI and how is it associated with different risk factors in adult surgery patients following hip and knee arthroplasties, compared to patients without a CDI diagnosis in the 30 days following the surgery?

## 2. Materials and Methods

Retrospective records for all patients who underwent hip or knee arthroplasties in 2017 were extracted from the Polish NHF (the only public payer in Poland) databases. The data include only procedures performed within the general health care system. The current analysis employs data on adults who underwent hip or knee prosthesis surgery, identified on the basis of the following ICD-9 codes: 81.51–81.55, 00.7 or 00.8. We excluded surgeries performed in people under 18 years old (n = 58). In total, 83,525 cases of hip (HPRO) and knee (KPRO) arthroplasties were included in the present analysis. The inclusion criteria were (1) hip or knee prosthesis reimbursed by the NHF and performed between 1 January 2017 and 31 December 2017, and (2) patient age (18 or more years old). The studied population has previously been described in detail [7].

We used the NHF data on the 192 CDI observed in the 30 day post-discharge period (hospitalizations, outpatient visits, etc.). We analyzed the demographic (age, place of residence) and clinical (chronic comorbidities, operated joint: hip/knee) characteristics of the patients, the number of drug groups prescribed, whether it was a primary or revision surgery according to the appropriate ICD-9 code, length of hospital stay (LOS), whether there was a stay in the intensive care unit (ICU), cause of surgery, postoperative infections, primary/revision surgery, as well as the long-term care after surgery. Patients with these particular disease entities at the preoperative stage were assessed based on the use of specific drug groups during a period of one year before the date of the procedure. The criterion for polytherapy was taking drugs from 5 or more ATC-code groups. Only prescribed drugs were taken into consideration.

HCA-CDI were diagnosed (with or without microbiological confirmation) and classified based on the uniform definitions issued by the ECDC, in accordance with the decision of the European Commission 2002/253/EC laying down case definitions for reporting communicable diseases to the community network [8]. According to the ECDC, the origin of a CDI case can be defined as healthcare-associated (HCA-CDI) based on the date (within 30 days of discharge from a healthcare facility) and location of the onset of CDI symptoms [9]. A 30 day time span was also employed in relation to post-discharge antibiotic prescription. The sensitivity of post-discharge CDI surveillance and postoperative follow-up of patients was dependent on the primary care physician and the relationship between inpatient and outpatient ABX treatment is unknown.

The post-surgery patient-days were calculated as follows: the number of days between the operation day and the day of the outpatient visit due to CDI or hospitalization due to CDI during the 30 day post-operation period.

Antibiotic prescriptions were expressed per 1000 surgery patients with an antibiotic prescription on discharge from hospital, or in ambulatory care, respectively in the 30 day post-surgery period using the ATC (Anatomical Therapeutic Chemical) system of the World Health Organization [10]. Only antibiotics for systemic use were taken into account—no antifungal (J02), antimycobacterial (J04), or antiviral (J05) drugs were included in the analyses. Combination antibiotic therapy was defined as the prescription of two or more different antibiotics.

This work was approved by the Bioethics Committee of Jagiellonian University (approval no.1072.6120.149.2020). The study was based on data gathered during routine patient care and the analysis did not include any individual participant’s data.

## 3. Statistical Analysis

The incidence of CDI (per 1000 patients) was calculated along with exact binomial 95% confidence intervals (95%CI), and compared in relation to demographic characteristics, and the description of the hospitalization index (due to knee or hip arthroplasty), as well as antibiotic prescriptions in post-hospital period by a Chi-squared test. In addition, multivariable logistic regression was applied to assess the strength of association between CDI and the chosen risk factors. The independent variables were chosen based on both (a) the significance of difference in CDI incidence, and (b) factors that can be related to CDI incidence based on literature searching and experience. The results are presented as odds ratios (OR), together with 95% confidence intervals (95% CI). Results with a *p*-value < 0.05 were considered statistically significant. All analyses were performed in IBM SPSS Statistics 26 (PS IMAGO PRO, Predictive Solutions Sp. z o.o.).

## 4. Results

In total, there were 192 CDIs, including 160 cases after HPRO (0.3/1000 patients) and 32 after KPRO (0.1/1000 patients). The CDI incidence rate was 34.4 per 10,000 patients, and 7.7 cases per 100,000 post-surgery patient-days (pds).

Patients who required Combination ABX therapy in the postoperative period suffered from CDIs significantly more often than patients who did not undergo such therapy, as follows: 2.8/1000 patients vs. 1.8/1000 patients (Table 1). A similar result was found in patients operated on due to trauma, in whom 7.9/1000 patients experienced CDIs, compared with those operated on due to degeneration (1.2/1000 patients) or complication (3.2/1000 patients); most often, CDI concerned a small group of patients not qualified elsewhere (reason: others, 6.7/1000 patients). However, in patients admitted in an emergency, post-discharge CDI was diagnosed significantly more often (4.6/1000 patients) than in patients admitted for scheduled procedures (1.7/1000 patients) (Table 1).

The median stay in hospital for CDI cases was 7 days. This result is in the third quartile of patients without CDI. Significantly more often, CDI occurred in patients from a small group requiring hospitalization in the ICU, as follows: 44.8/1000 patients vs. 2.1/1000 patients among non-ICU patients (Table 1). Also, patients who required long-term care after discharge were infected more often than patients who did not receive such care, as follows: 15.6/1000 vs. 2.2/1000 patients (Table 1).

A total of 3531 people received a prescription of antibiotics on discharge. The incidence in this group was 3.1/1000 patients and was not significantly higher than for the patients who were not prescribed an antibiotic (2.3/1000) (Table 1).

Significantly more frequently, CDI occurred in patients with other healthcare-associated infections in the post-discharge period; the incidence rate in this group was 27/1000 patients. Of all studied surgical patients, 763 were prescribed at least one antibiotic in outpatient care in the 30 day postoperative period (Table 2), and in these patients, more often than in patients without antibiotic treatment, CDI was diagnosed (55.0/1000 patients vs. 1.8/1000 patients) (Table 2).The incidence of CDI increased with the number of prescriptions filled; the incidence with one prescription was 53.9/1000 patients, and with more than one prescription, 57.8 /1000 patients (Table 2).

The incidence of post-discharge CDI associated with antibiotic treatment was significantly dependent on the class of the antibiotic; the highest incidence was associated with other beta-lactams (97.0/1000 patients), beta-lactams, penicillins (58.8/1000 patients) and quinolones (45.5/1000 patients) (Table 2).

In the multifactorial analysis, the following factors were found to be significant: age over 65 years of age, stay in the intensive care unit, trauma as the cause of surgery, length of stay over 7 days, HAIs other than CDI and taking beta-lactams and other beta-lactams and/or quinolones J01M, but not macrolides, in the post-discharge period (Table 3).

## 5. Discussion

This study demonstrates that the risk of the occurrence of CDI is closely associated with both factors independent of the course of medical care, such as age or need for surgery due to an injury, and factors directly related to the healthcare system, such as the need to treat HAIs or the use of antibiotics in such patients.

The fact that the risk of CDI increases with age and with deteriorating general condition, when patients require long-term care, has been confirmed by other authors [11,12]. Antimicrobials are some of the most commonly prescribed medications in long-term care settings. Therefore, it is not surprising that *Clostridium difficile* associated diarrhea is common in older residents. CDI has been identified as the most frequent cause of non-epidemic acute diarrhea in long-term care facilities (LTCF) [13]. In our study, people over the age of 65 were diagnosed with CDI three times more often than younger people, and patients requiring long-term care following surgery were eight times more prone to it than people who did not require such care. Additionally, people who underwent combination antibiotic therapy suffered from CDI twice as often than patients who did not undergo polytherapy. Patients who requires constant care often suffer from many diseases and usually have a weakened immune system, altered gut microbiota or other related severe pathologies. The study by Donskey et al. suggests that asymptomatic carriage and CDI among the LTCF residents contribute to the transmission of *C. difficile* in both LTCF and in hospital during admissions to the unit. There is a need for greater emphasis on infection control measures and the management of antimicrobial drugs in LTCF [14].

A study by Weber et al. demonstrated that antibiotics are prescribed more often to LTCF patients, which may be associated with an increased risk of poor outcomes in the post-discharge period [15]. Also, insufficient post-discharge care caused by poor communication with patients and/or improper conveying of information between hospital medical staff and caregivers may result in an increased probability of adverse events and pre-hospitalization [16].

Perioperative proceedings, especially a prolonged stay and the need for an ICU stay, were found to be of great significance. The percentage of people with CDI increased with prolonged hospital stays; similar results were obtained by Hung et al. [17]. CDI is one of the most important causes of hospital-acquired infections in intensive care units (ICU) [18], which may be related to the amount of antibiotics prescribed there; approximately 70% of ICU patients are prescribed antibiotics [19]. Also, increasing duration of antimicrobial prophylaxis is associated with higher odds of CDI. These findings highlight the idea that stewardship efforts to limit the duration of prophylaxis have the potential to reduce adverse events without increasing SSI [20]. A similar phenomenon has been observed by the authors with regard to another patient population, i.e., subjects requiring urogenital procedures, in which, on the basis of regression coefficients, a positive correlation was demonstrated between the use of fluoroquinolones and the HA-CDI incidence rate [5].

The relationship between the class of antibiotics prescribed and the incidence of CDI has been debated for several years. The antibiotics associated with the highest risk of developing CDI include the following: clindamycin, fluoroquinolones, cephalosporins, and carbapenems [21]. Knecht et al. showed that different antibiotic classes (cephalosporins, ampicillin/sulbactam and quinolones) affect the gut microbiota similarly at the RNA level but differently at the DNA level, and this explains why several different antibiotics are associated with similar intestinal problems [22].

However, a significant outcome of our study was the fact that the need for antibiotic treatment of infections is a significant risk of CDI, as well as the occurrence of another infection.

This observation has practical implications for infection prevention and control in Poland and is consistent with the results of other studies in which higher rates of incidence of surgical site infections were observed following operations associated with implantation of both knee and hip joints. Ziółkowski et al. found incidences of SSIs after hip and knee arthroplasty surgeries of 5.8% and 5.4%, respectively (2013–2015), while the average rates reported in the HAI-Net SSI ECDC program for 2015 were 1.1% and 0.6%, respectively [23,24].

Observation of risk factors for infection has been carried out in over 39,000 patients undergoing primary hip or knee arthroplasty; the data came from a large database of the American College of Surgeons National Surgical Quality Improvement Program [25]. In this study, the frequency of CDI incidents in the 30 day postoperative period was 0.10% (95% CI, 0.07–0.13%). The independent preoperative and procedural risk factors were older age, functional dependence, preoperative anemia, hypertension, and hip arthroplasty. Postoperative risk factors included urinary tract infection, sepsis, and the development of any infection.

For many years, a relationship has been known to exist between a reduction in gastric acidity and the risk of *Clostridioides difficile* infection and the development of pseudomembranous enterocolitis, especially with concurrent antibiotic therapy. This relationship is visible with histamine 2 receptor antagonist treatment, but with proton pump inhibitors (PPIs), this association is much greater [26]. A meta-analysis of 23 studies involving approximately 300,000 patients demonstrated that PPI treatment increases the risk of developing this infection by nearly 65% [27]. Modification of the gut microbiota composition, associated with PPI treatment, also promotes the development of small intestinal bacterial overgrowth syndrome (SIBO). A meta-analysis of 19 studies encompassing over 7000 patients found that this risk grows by around 70% [28]. Of other drugs of potential relevance to CDI infection, a meta-analysis by Furuya-Kanamori et al. demonstrated the importance of exposure to corticosteroids [29]. Unfortunately, the methodology of our study makes it impossible to verify the importance of the application of particular groups of drugs for the occurrence of CDI.

Our results have several important implications for clinical practice in Poland, especially for the area of infection control. Multimorbidity in the preoperative period and beta-lactams or quinolones prescription within 30 days of discharge from hospitals are associated with increased CDI rates. Thus, our observation confirms the necessity of implementing effective antimicrobial stewardship programs in Polish hospitals, as well as continuous education for physicians working in the community sector, since according to several recent studies on antimicrobial consumption in Poland this is not optimal [30]. Additionally, another HAI–other than CDI–in the post-discharge period is associated with a higher risk of CDI, which is directly related to the ineffectiveness of infection control programs, lack of post-discharge surveillance in surgical patients, or difficulties with access to proper microbiological diagnostics in Polish hospitals. We have also shown that a stay in long-term care facilities after discharge is associated with higher CDI incidence, and the HALT-3 project revealed problems with infection control and antibiotic consumption in these settings in Poland [31].

The use of fluoroquinolones, due to their numerous side effects, should be limited to situations where there are no other therapeutic options available. Furthermore, according to a European Medicines Agency alert from 2018, in the elderly, quinolones should be used with special caution [32]. At present in Poland, quinolones are prescribed most frequently; they make up more than 40% of all antibiotic prescriptions. Unfortunately, there are concerns regarding the growth of fluoroquinolone prescriptions in Poland in recent years, as a percentage of the total antibiotic consumption [31].

There are also limitations to the presented results. Firstly, this was an observational study. The reason for prescribing antibiotics is unknown, as the available data relates only to the sale of drugs, not the consumption. Secondly, the level of CDI diagnostics is unknown (only clinical signs or performed diagnostic tests and their type). The role of proton pump inhibitors in the risk of CDI has not been analyzed, and information about perioperative antibiotic prophylaxis and its duration are unknown. In addition, the source of the data also makes it impossible to determine whether post-surgery inpatient and outpatient antibiotics were prescribed only for patients with confirmed infections. However, the value of our analysis is its pioneering nature, and the results indicate areas of surgical care that require more thorough research and intervention.

## 6. Conclusions

In Poland the CDI incidence rate is 34.4 cases per 10,000 patients, and 7.7 per 100,000 post-surgery pds. Multivariable analysis of risk factors for CDI up to 30 days after hospitalization shows that various factors influence the risk of developing CDI; among these factors are another healthcare-associated infection in the post-discharge period and antibiotic prescription within 30 days of discharge. These observations, together with the results of other studies in this area in Poland, indicate the necessity of improvement of infection control programs as the key factor for prevention of *C. difficile*, as well as continuous education for physicians working mainly in ambulatory care and the community.

## Figures and Tables

**Table 1 ijerph-19-03155-t001:** *Clostridioides difficile* infection incidence in relation to patient and operation characteristics.

Characteristics of the Study Group		Total, N = 83,525	No. of CDI Cases	Incidence (95% CI) Per 1000 Patients	*p*-Value
Sex	Female	52,624	129	2.5 (2.0–2.9)	0.229
	Male	30,901	63	2.0 (1.6–2.6)	
Age	<65	27,855	28	1.0 (0.7–1.5)	<0.001
	>=65 years	55,670	164	2.9 (2.5–3.4)	
Place of residence–city *	Yes	48,583	111	2.3 (1.8–2.9)	0.944
	No	34,657	80	2.3 (1.9–2.8)	
Admission mode	Scheduled	66,222	112	1.2 (0.9–1.5)	<0.001
	Emergency	17,149	79	6.5 (5.4–7.9)	
Cause of surgery	Degeneration	66,227	79	1.2 (0.9–1.5)	<0.001
	Injury	11,967	94	7.9 (6.4–9.6)	
	Complications	4736	15	3.2 (1.8–5.2)	
	Other	595	4	6.7 (1.8–17.1)	
Joint operated on	Hip	56,068	160	2.9 (2.4–3.3)	<0.001
	Knee	27,457	32	1.2 (0.8–1.6)	
Surgery	Primary	78,388	175	2.2 (1.9–2.6)	0.119
	Revision	5137	17	3.3 (1.9–5.3)	
ICU during the 1st hospitalization	No	83,056	171	2.1 (1.8–2.4)	<0.001
	Yes	469	21	44.8 (27.9–67.6)	
Prolonged hospitalization *	No	66,009	119	1.8 (1.5–2.2)	<0.001
	Yes	17,501	72	4.1 (3.2–5.2)	
ABXs on discharge	No	79,994	181	2.3 (1.9–2.6)	0.301
	Yes	3531	11	3.1 (1.6–5.6)	
Combination ABX therapy **	No	40,214	71	1.8 (1.4–2.2)	0.002
	Yes	43,311	121	2.8 (2.3–3.3)	
Long-term postoperative care	No	82,818	181	2.2 (1.9–2.5)	<0.001
	Yes	707	11	15.6 (7.8–27.7)	

* more than the third quartile of stay, no data for 285 patients; ** Combination antibiotic therapy was defined as the prescription of two or more different antibiotics; ABX—antibiotic; ICU—intensive care unit.

**Table 2 ijerph-19-03155-t002:** Antibiotic prescriptions in the post-discharge period in ambulatory care.

Characteristics of the Study Group		Total, N= 83,525 *	Number of Cases	Incidence (95% CI) Per 1000 Patients	*p*-Value
ABX prescriptions	No	82,762	150	1.8 (1.5–2.1)	
	Yes	763	42	55.0 (40.0–73.7)	
Non-CDI HAI	No	81,765	144	1.8 (1.5–2.1)	<0.001
	Yes	1760	48	27.3 (20.2–36.0)	
Number of ABX prescriptions	No	82,762	150	1.8 (1.5–2.1)	<0.001
	1	538	29	53.9 (36.4–76.5)	
	>1	225	13	57.8 (31.1–96.8)	
J01C, beta-lactams, penicillins	No	83,320	180	2.2 (1.9–2.5)	<0.001
	Yes	205	12	58.5 (30.6–100.0)	
J01D, other beta-lactams	No	83,391	179	2.1 (1.8–2.5)	<0.001
	Yes	134	13	97 (52.7–160.2)	
J01F, macrolides	No	83,424	188	2.3 (1.9–2.6)	<0.001
	Yes	101	4	39.6 (10.9–98.3)	
J01M, quinolones	No	83,217	178	2.1 (1.8–2.5)	<0.001
	Yes	308	14	45.5 (25.1–75.1)	
J01, others	No	83,365	184	2.2 (1.9–2.5)	<0.001
	Yes	160	8	50 (21.8–96.1)	

* no data for 15 patients, ABX—antibiotic, CDI—*Clostridioides difficile* infections, HAI—healthcare associated infections. J01, others antibacterial for systematic use.

**Table 3 ijerph-19-03155-t003:** Multivariable analysis of the influence of the studied factors on the incidence of CDI.

Study Group	OR	95%CI	*p*
Man	0.98	0.71–1.34	0.901
Age > 65 years	1.78	1.16–2.73	0.008
multimorbidity	1.05	0.72–1.55	0.791
J01C, beta-lactams, penicillins	4.18	2.04–8.57	<0.001
J01D, other beta-lactams	6.41	3.15–13.02	<0.001
J01F, macrolides	2.69	0.88–8.24	0.084
J01M, quinolones	2.80	1.42–5.54	0.003
J01, other antibacterial for systematic use	3.17	1.39–7.25	0.006
Long-term care after discharge	2.55	1.32–4.93	0.005
Knee surgery	0.82	0.53–1.27	0.374
ICU	13.17	7.95–21.81	<0.001
Revision	0.85	0.23–3.14	0.810
HAI (not with *C. difficile*)	3.58	2.15–5.94	<0.001
Cause of surgery, degeneration (ref.)	Ref		
other	2.53	0.77–8.35	0.128
complications	1.71	0.43–6.89	0.448
injury	3.44	2.37–4.98	<0.001
Hospitalization for more than 7 days	1.62	1.19–2.2	0.002

OR—odds ratio, 95%CI—95% confidence interval, *p*—probability value, ICU—intensive care unit, HAI—healthcare associated infections.

## Data Availability

The datasets during and/or analysed during the current study available from Estera Jachowicz (e-mail: estera.jachowicz@doctoral.uj.edu.pl) on reasonable request.

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
