# Peer review of "Post-Discharge Clostridioides difficile Infection after Arthroplasties in Poland, Infection Prevention and Control as the Key Element of Prevention of C. difficile Infections"

_ijerph, 2022, doi:10.3390/ijerph19063155_

Round 1

Reviewer 1 Report

Thank you for the opportunity to review this article. The article investigated the risk factors for clostridium difficile infection after hip or knee arthroplasty in Poland. The article is well-written and the conclusions are consistent with the findings. In view of the limited evidence on this topic in the literature, I think the article is suitable for publication. However, I have a couple of considerations which need to be addressed:

- In table 2, I would like the significance to be made explicit in addition to the confidence interval.
- Regarding the multivariate analysis, how were the variables included chosen? I would like this to be made explicit in the methods section.

Author Response

Thank you for the opportunity to review this article. The article investigated the risk factors for clostridium difficile infection after hip or knee arthroplasty in Poland. The article is well-written and the conclusions are consistent with the findings. In view of the limited evidence on this topic in the literature, I think the article is suitable for publication.

Authors’ reply: Thank you for this comment!

However, I have a couple of considerations which need to be addressed:

- In table 2, I would like the significance to be made explicit in addition to the confidence interval.

Authors’ reply: Corrected according to suggestions. The table 2 was supplemented with significance.

- Regarding the multivariate analysis, how were the variables included chosen? I would like this to be made explicit in the methods section.

Authors’ reply: Corrected according to suggestions. The independent variables were chosen based on both a) the significance of difference in CDI incidence as well as b) factors that can be related to CDI incidence based on literature search and experience (Lines 115-117)

Reviewer 2 Report

Specific comments

Abstract

Line 20. Please change "Patients" to "Methods and Patients".

Line 18. Please the sentence: "CDI and/or antibiotic prescriptions in the 30-day post-surgery period were expressed per 1000 surgeries with antibiotic prescription on discharge or in ambulatory care, respectively, should be placed after the sentence beginning with "  Retrospective records for 83.525 surgery patients,..."

The conclusion of the abstract should be somewhat shorter. Perhaps a sentence like "postoperative antibiotic prescription in patients undergoing joint replacement surgery is the main risk factor for CDI."

Introduction.

The opening sentence of the introduction should be modified and clarify that CDI is the most frequent cause of nosocomial diarrhea. I would not mention anything about antibiotic resistance because it is not related to the content of the research. In the first two paragraphs, some additional bibliographic reference that is related to the information presented should be introduced.

Line 58. The abbreviation of NHF (Polish National Health Fund) should be defined the first time it appears in the manuscript, which is line 58.

Material and method.

More information is needed on postoperative follow-up of patients. Is it possible for a patient to present CDI and not be detected by the system?. Is there a connection or communication between the patient record and the comments of the primary care physician?. It should be specified in this section whether the antibiotic prescribed as antibiotic prophylaxis and its duration have been taken into account.

Results

Line 119. I think it is relevant to report whether the difference between the incidence of hip and knee prostheses is statistically significant or not. According to the calculations I have made, it could be significant. If such a difference is demonstrated, some comment should be made in the discussion, even though it seems that the multivariate analysis does not confirm this possible result. 

Line 121. Please define "combination therapy" (I understand you mean the prescription of two different antibiotics).

Line 134. This sentence should be changed for better understanding.

Line 139. It should be clear that the “prescription” refers to antibiotics.

Has the role of proton pump inhibitors in the risk of CDI been analyzed?. If no, it should be commented in the limitations paragraph because it could the fact of not analyzing it could affect the association found with other variables (age, admission to the ICU) and the risk of DIC. Some information on antibiotic prophylaxis and its duration is also missing. If it has not been taken into account in the study, it should also be considered a limitation.

Discussion

Line 177. It should also be specified that combined treatment refers to combined antibiotic treatment.

Line 210. I don't quite understand this sentence. What other factor, besides antibiotic treatment, can influence the risk of CDI?

Line 220. The paragraph beginning on this line does not refer to the results of this research and should therefore be considered for deletion.

Line 259. All comments about the high incidence of bacteremia in Poland could be removed because they have not been an objective of the study.

Table 1. I do not understand how there is a significant difference in two complementary groups such as "primary" and "Revision".

Author Response

Authors’ reply: Thank you for your time reviewing our manuscript, valuable feedback and favorable reception. Our answers below:

Specific comments, Abstract

Line 20. Please change "Patients" to "Methods and Patients".

Line 18. Please the sentence: "CDI and/or antibiotic prescriptions in the 30-day post-surgery period were expressed per 1000 surgeries with antibiotic prescription on discharge or in ambulatory care, respectively, should be placed after the sentence beginning with "  Retrospective records for 83.525 surgery patients,..."

The conclusion of the abstract should be somewhat shorter. Perhaps a sentence like "postoperative antibiotic prescription in patients undergoing joint replacement surgery is the main risk factor for CDI."

Authors’ reply: Corrected according to suggestions.

Introduction. The opening sentence of the introduction should be modified and clarify that CDI is the most frequent cause of nosocomial diarrhea. I would not mention anything about antibiotic resistance because it is not related to the content of the research. In the first two paragraphs, some additional bibliographic reference that is related to the information presented should be introduced.

Line 58. The abbreviation of NHF (Polish National Health Fund) should be defined the first time it appears in the manuscript, which is line 58.

Authors’ reply: Corrected according to suggestions. We changed “Introduction” section and added additional bibliographic reference.

Material and method. More information is needed on postoperative follow-up of patients. Is it possible for a patient to present CDI and not be detected by the system?. Is there a connection or communication between the patient record and the comments of the primary care physician?. It should be specified in this section whether the antibiotic prescribed as antibiotic prophylaxis and its duration have been taken into account.

Authors’ reply: Corrected according to suggestions. The “Materials and Methods” section (lines 95-97) and “limitations” (lines 291-295) were supplemented

Results

Line 119. I think it is relevant to report whether the difference between the incidence of hip and knee prostheses is statistically significant or not. According to the calculations I have made, it could be significant. If such a difference is demonstrated, some comment should be made in the discussion, even though it seems that the multivariate analysis does not confirm this possible result. 

Authors’ reply: Corrected according to suggestions. ….

Line 121. Please define "combination therapy" (I understand you mean the prescription of two different antibiotics).

Authors’ reply: Corrected according to suggestions. We added a define (line …) “The combination antibiotic therapy, was defined as the prescription of two or more different antibiotics”, and changed “combination therapy“ on “combination antibiotic therapy” (line …)

Line 134. This sentence should be changed for better understanding.

Authors’ reply: Corrected according to suggestions, incorrectly entered numerical values have been corrected.

Line 139. It should be clear that the “prescription” refers to antibiotics.

Authors’ reply: Corrected according to suggestions.

Has the role of proton pump inhibitors in the risk of CDI been analyzed?. If no, it should be commented in the limitations paragraph because it could the fact of not analyzing it could affect the association found with other variables (age, admission to the ICU) and the risk of DIC. Some information on antibiotic prophylaxis and its duration is also missing. If it has not been taken into account in the study, it should also be considered a limitation.

Authors’ reply: Corrected according to suggestions. “Limitations” section was supplemented.

Discussion Line 177. It should also be specified that combined treatment refers to combined antibiotic treatment.

Authors’ reply: Corrected according to suggestions, “the combination antibiotic therapy” was defined in “Materials and Methods” section.

Line 210. I don't quite understand this sentence. What other factor, besides antibiotic treatment, can influence the risk of CDI?

Authors’ reply: Corrected according to suggestions.

Line 220. The paragraph beginning on this line does not refer to the results of this research and should therefore be considered for deletion.

Authors’ reply: Corrected according to suggestions, the paragraph has been deleted.

Line 259. All comments about the high incidence of bacteremia in Poland could be removed because they have not been an objective of the study.

Authors’ reply: Corrected according to suggestions.

Table 1. I do not understand how there is a significant difference in two complementary groups such as "primary" and "Revision".

Authors’ reply: Corrected according to suggestions. The table 1 was supplemented with significance.

Round 2

Reviewer 2 Report

I recommend that the article be accepted in its present form.